# Impact of Long-Lasting Environmental Factors on Regulation Mediated by the miR-34 Family

**DOI:** 10.3390/biomedicines12020424

**Published:** 2024-02-12

**Authors:** Peter Štefánik, Martina Morová, Iveta Herichová

**Affiliations:** Department of Animal Physiology and Ethology, Faculty of Natural Sciences, Comenius University in Bratislava, Ilkovičova 6, 84215 Bratislava, Slovakia; peter.stefanik@uniba.sk (P.Š.); morova4@uniba.sk (M.M.)

**Keywords:** phthalates, electromagnetic, circadian, *cry1*, cancer, neural

## Abstract

The present review focuses on the interactions of newly emerging environmental factors with miRNA-mediated regulation. In particular, we draw attention to the effects of phthalates, electromagnetic fields (EMFs) and a disrupted light/dark cycle. miRNAs are small non-coding RNA molecules with a tremendous regulatory impact, which is usually executed via gene expression inhibition. To address the capacity of environmental factors to influence miRNA-mediated regulation, the miR-34 family was selected for its well-described oncostatic and neuro-modulatory properties. The expression of miR-34 is in a tissue-dependent manner to some extent under the control of the circadian system. There is experimental evidence implicating that phthalates, EMFs and the circadian system interact with the miR-34 family, in both lines of its physiological functioning. The inhibition of miR-34 expression in response to phthalates, EMFs and light contamination has been described in cancer tissue and cell lines and was associated with a decline in oncostatic miR-34a signalling (decrease in *p21* expression) and a promotion of tumorigenesis (increases in *Noth1*, *cyclin D1* and *cry1* expressions). The effects of miR-34 on neural functions have also been influenced by phthalates, EMFs and a disrupted light/dark cycle. Environmental factors shifted the effects of miR-34 from beneficial to the promotion of neurodegeneration and decreased cognition. Moreover, the apoptogenic capacity of miR-34 induced via phthalate administration in the testes has been shown to negatively influence germ cell proliferation. To conclude, as the oncostatic and positive neuromodulatory functions of the miR-34 family can be strongly influenced by environmental factors, their interactions should be taken into consideration in translational medicine.

## 1. Introduction

Nowadays, the human population lives in a diametrically different environment from that of our ancestors only one century ago. Most of the population is concentrated in cities. Artificial lighting blurs the distinction between day and night. Our food, beverages and cosmetics are wrapped in disposable packaging materials derived from crude oil containing endocrine disruptors. We spend a significant part of the day in contact with a pocket computer and Wi-Fi routers that, during their operation, generate electromagnetic radiation at a strength and frequency at which the long chain of our ancestors was not exposed at all. 

These artificial stressors in urban areas are relatively weak, and humans must be exposed to them for a long time before a physiological effect is observed. This is part of the problem when approaching this issue experimentally, as research studies focused on the influence of the above-mentioned factors usually have to be designed in such a way that the tested effect can be analysed before decades of exposure have occurred. Moreover, details of the experimental design differ between specific environmental factors. The capacity of epidemiological studies to address the problem of single and/or cumulative influences of newly emerged environmental factors present in urban areas is also limited. Moreover, high variability in the obtained data complicates the interpretation of the results. Therefore, research focused on combined effects of the environmental factors influencing human health is rare.

Recently, a new class of signal molecules, the small non-coding miRNAs (miRNAs), were discovered, and their regulatory impact on the organism was observed at the level of all organ systems. We suppose that the functioning of miRNAs can be influenced by external environmental factors, and in this way, they can mediate the impact of environmental conditions on the organismal milieu.

Therefore, the principal aims of the present review are to collect, analyse and synthesise available scientific resources on the effects of endocrine disruptors and electromagnetic and light pollution on miRNA expression and correlate them with particular physiological functions. As the light (L)/dark (D) cycle is a major synchronizing factor of the circadian system regulating biological rhythms with a 24 h period, we also focused on the impact of circadian rhythm disruption on miRNA expression. 

To manifest possible synergic effects of environmental factors on miRNA-mediated regulation, the miR-34 family was selected, as these miRNAs have been an experimental focus of our research group for several years; more importantly, miR-34a was the first miRNA to be clinically tested for cancer treatment and did not produce the anticipated results [1]. We suppose that a broader description of miRNA functioning must be available before its use in translational medicine, and besides individual variability and accompanying comorbidities with respect to the aim of treatment, the impact of environmental factors on miRNAs should also be considered.

## 2. Functions of Small Non-Coding RNAs and Their Modulation by Environmental Factors

Small non-coding RNAs (miRNAs) are a large family of RNA molecules approximately 19–25 nucleotides long that regulate intracellular processes directly, without serving as a template for protein synthesis. The regulatory influence of miRNAs relies on interference with the translation and/or transcription of protein-coding genes, depending on their sequence. In most cases, miRNAs cause an inhibition of gene expression [2]. More than 2000 human miRNA species occupying 1–5% of the genomic DNA have been annotated in the database miRBase. Therefore, it is not surprising that miRNAs influence up to 90% of protein-coding genes in humans [3,4,5] and impact many physiological processes, including development; cell cycle progression; endocrine, cardiovascular and immune system functioning; metabolism; and others [6,7,8,9,10].

miRNA transcription begins in the nucleus, employing DNA as a template; in this respect, it resembles mRNA transcription. In later steps, however, miRNA biogenesis continues differently and is more complex than mRNA expression. miRNAs localised in the introns of coding genes can be transcribed together with their host genes; however, sequences of intergenic miRNA can contain miRNA-specific regulatory elements [11,12,13]. In both cases, miRNAs are transcribed, usually by RNA polymerase II, to produce a primary transcript of miRNA (pri-miRNA) containing a structure resembling a hairpin. In the next step, most of the 5′ and 3′ ends of the hanging sequences of pri-miRNA hairpins are cut off by a complex composed of RNase III Drosha and DGCR8 (DiGeorge syndrome critical region gene 8 protein) to create a precursor miRNA (pre-miRNA). After the translocation of pre-miRNAs into the cytoplasm, RNase III Dicer removes the terminal loop of the hairpin to produce the mature form of miRNA consisting of two short antiparallel, partly complementary RNA sequences [2]. 

The execution of miRNA function depends on its incorporation into the RISC complex (RNA-induced silencing complex), whose key component is a member of the Argonaut protein family (AGO), usually AGO2. Based on the sequence of the mature miRNA, one of its strands (guide strand) is loaded into the RISC. The fate of the second strand of miRNA (passenger strand) depends on the biological context, but its level in the cell is usually much lower than that of the guide strand [14]. 

The miRNA-induced silencing of target gene transcription is based on the interference of the ‘seed sequence’ of miRNA with the complementary miRNA response element (MRE) of mRNA. In most cases, the MRE is located in the 3′ untranslated region (UTR); however, miRNA interference with MRE located in the 5′UTR and promoter has been described as well. The interaction of miRNA with MRE is guided by the RISC complex. Depending on the degree of complementarity between the seed sequence and MRE, the mRNA of the target gene can be degraded by AGO2, deadenylated and consequently degraded, and/or the inhibition of translation can take place [15]. mRNA degradation by AGO2 occurs when the miRNA seed sequence is fully complementary to the MRE region. However, there is usually at least one mismatch in the miRNA and MRE sequences. In the case of imperfect complementarity, the mRNA of the host gene is first deadenylated, translation is then inhibited, and the mRNA is consequently degraded [16]. 

miRNA usually targets more than one mRNA and, reciprocally, the expression of one gene is in most cases regulated by more than one miRNA. Although exact numbers are not known, it is estimated that miRNA can interfere with 100–200 targets [9,17,18], which implicates more than 2 million possible combinations in the case of the human genome. On the other hand, it was shown that far fewer miRNA–mRNA binding events can result in the effective modification of some intracellular events [19]. The database of validated miRNA-mediated silencing, miRTarBase, at present registers more than 300 000 miRNA–mRNA interactions [20]. miRNA-induced silencing seems to be strongly tissue-specific [21].

miRNAs usually execute regulation inside the cell where they were synthesised, but they also can be released into the intracellular space or circulation and influence the cells and/or tissues that take them up. The spectrum of circulating miRNA has been associated with several pathologies, e.g., tumorigenesis, hepatitis C virus (HVC), COVID-19, osteoporosis, heart failure and infarction, diabetes and others [7,22]. There are several ways in which miRNAs can be released into the blood stream; however, those that are packed into exosomes are protected from degradation and are most likely to provide information about changes in the tissue of their origin. Therefore, the measurement of exosomal miRNAs in human fluids, which is called “liquid biopsy”, is at present extensively tested in translational medicine and in some cases already available for use to determine or refine diagnosis [23,24]. The therapeutic potential of several miRNAs has been tested in clinical studies focused on cancer, HVC, heart failure, Crohn’s disease, ulcerative colitis, amyotrophic lateral sclerosis and some others [2,23,24].

In spite of obvious progress in employing silencing in translational medicine, the potential utility of miRNA is limited by several factors, including off-target activity. Off-target activity results from a short seed sequence (6–7 nucleotides) that allows miRNA–mRNA complementarity-based binding with more than one gene [2,4]. Therefore, further research is needed to improve the ratio of successful clinical studies focused on silencing. 

miRNAs with a strong functional connection to a cell cycle regulator are frequently investigated for their capacity to influence tumorigenesis. Among them, the oncostatic capacity has been attributed to miR-34a [25,26]. miR-34a and the tumour suppressor p53 create a positive regulatory feedback loop. The expression of miR-34a is induced by p53 via a p53-binding domain in the promoter of the miR-34a coding gene [27], and miR-34a facilitates p53′s effects by increasing its stability through the inhibition of negative p53 regulators Mdm4 (Figure 1; [28]) and Sirt1 [29].

Research is still ongoing to improve the accessibility of miR-34 family members in translational research and clinical practice in respect to the treatment of breast cancer [41], colon cancer [42], lung cancer [43], oral cancer [44], oesophageal cancer [45] and some others (reviewed in [46]).

Given its functional relationship with p53 and therapeutic potential, miR-34a was among the first miRNA for which its effects were clinically tested for the treatment of solid tumours; however, these clinical trials did not produce the expected results [1,47]. Studies that indicated miR-34a as a tumour suppressor have been criticised for the use of supra-physiological levels of synthetic mimics [48], and the off-target activity of miR-34a has been demonstrated recently [37]. We have also observed the effect of a high-frequency electromagnetic field (RF-EMF) on the oncostatic functions of miR-34a [36].

Despite the above-mentioned challenges, research on the therapeutic potential of miR-34a still continues. The aim is to decrease the off-target effects and toxicity of treatment and increase miR-34a stability. Abdelaal et al. [49] developed a chemically modified miR-34a mimic (FM-miR-34a) that has been evaluated under in vitro and in vivo conditions. The administration of FM-miR-34a caused inhibition of cancer cell proliferation, migration and invasion. The inhibitory effect of FM-miR-34a on tumour growth was observed in tumour-bearing mice, resulting in a clear tumour-static effect. Treatment was not accompanied by an immune response. These results indicate the enhanced stability and activity of FM-miR-34a compared to those of the former miR-34a mimic delivery system. Nonetheless, future studies are needed to ensure its safety and efficacy in humans.

As it looks like miR-34a can integrate the signalling of several regulatory factors to the internal milieu, we focused on the possibility that environmental factors can modify miRNAs signalling. In particular, we brought attention to the effects of phthalates, EMFs and the LD cycle in the context of the circadian system on the levels of miR-34 family members and their functioning.

The miR-34 family consists of miR-34a, miR-34b and miR-34c. Genes encoding miR-34a and miR-34b/c are located on chromosomes 1 and 11, respectively. The seed sequences of mir-34a and miR-34c are identical, and the seed sequence of miR-34b is very similar to them [26]. All three members of the miR-34 family are considered tumour suppressors; their transcription is induced by p53 and inhibited by the zinc finger proteins SNAIL and ZEB1 and the cytokine TGF-β [50,51].

### 2.1. Interference of Phthalates on the Functions Executed by miR-34

Phthalates are a family of ubiquitous environmental pollutants that are produced in large quantities to provide flexibility and durability to plastic materials. They are also used to improve the lubrication of other substances and can be utilised as solvents. Phthalates can be found in a wide range of products used in daily life, and since there are no covalent bonds between phthalates and the plastics in which they are mixed, they can leach from these products into the environment and enter the body through food consumption, inhalation, dermal contact or intravenous injection [52]. 

Phthalates can act as endocrine disruptors [53] or they can cause DNA damage through the induction of reactive oxygen species (ROS) generation [54]. Phthalate uptake is associated with abnormalities in reproductive tract development [55,56], brain development [57] and behavioural changes [58,59] and can promote carcinogenesis [60,61]. When acting as endocrine disruptors, phthalates bind to oestrogen receptors [62], androgen receptors [63], thyroid receptors [64] and human peroxisome proliferator-activated receptors [65] and modify their roles in the maintenance of homeostasis and the regulation of developmental processes [66]. In addition to direct interactions with receptors, phthalates can affect the functioning of the endocrine system by influencing hormone biosynthesis [67,68], transport [68,69] and metabolism [68,70]. Another way in which phthalates can modulate physiological processes is through their interference with miRNAs [56,71,72].

Endocrine-disrupting chemicals (EDCs) administered to mice during postnatal development up to day 60 caused a significant increase in the mRNA expression of Dicer, the enzyme involved in miRNA biosynthesis, in testes. EDCs also induced an increase in the expression of adenosine deaminase, contributing to the miRNA binding to the target gene and the miRNA-modifying enzyme Zcchc11. Similarly, histopathological and hormonal alterations were demonstrated in the testes of EDC-treated animals. EDC administration was associated with a deregulated expression of genes encoding proteins involved in steroidogenesis and a significant increase in the proportion of apoptotic germ cells. These effects were accompanied by pronounced changes in the mouse testicular miRNome [73], which is in accordance with remarkable changes in the miRNA expression induced by a phthalate mixture in the rat prostate [74]. 

In particular, up-regulation in the expression of two miRNAs, including miR-34b, and down-regulation in eight miRNAs were observed in EDC-treated mice. Both precursor and mature forms of miR-34b-5p were up-regulated in phthalate- and alkylphenol-exposed animals. Previously, it was shown that miR-34b promotes apoptosis [75]. Therefore, a rise in the miR-34b-5p expression can contribute to the observed increase in cell death in the testes after EDC exposure [73].

Interference between phthalates and miR-34a-mediated effects has also been demonstrated in tumorigenesis. With the use of two human prostate cancer cell lines, LNCaP and PC-3, it has been shown that the administration of butyl benzyl phthalate (BBP) strongly decreases the expression of miR-34a. The exposure of LNCaP and PC-3 cells to BBP for 6 days caused increase in the viability and cyclin D1 expression, while the expression of tumour suppressor p21 was inhibited. Ectopic miR-34a showed the opposite effects compared to BBP in both cell lines. Moreover, the effects of BBP were abolished when miR-34a and BBP were administered simultaneously [76]. The administration of dibutyl phthalate (DBP) and benzo[a]pyrene (BaP) caused a decrease in the miR-34a expression and an increase in Notch signalling in the rat liver [77]. Notch signalling is associated with cell inflammation and cancer progression [78]. Accordingly, phthalate administration was accompanied by an increase in the serum levels of pro-inflammatory cytokines IL-1β and iNOS, while serum concentrations of the anti-inflammatory factors TGF-β and CCL22 decreased. In 293-T cells, miR-34a administration caused an inhibition of the *Notch1* expression, providing a possible explanation for the up-regulation in Notch signalling under conditions of phthalate exposure and a decrease in the miR-34a levels [77]. These results are in accordance with the accepted opinion that miR-34a is a tumour suppressor [25,26], while phthalates exert the opposite effects [79,80]. 

In the context of phthalate intoxication, a concern was also raised regarding female reproductive health, as phthalate exposure was associated with an increased incidence of *uterine leimyoma* [81]. A preliminary study focused on the association between exposure to phthalates and the miRNA expression in fibroid uterine tumours showed that the expression of several miRNAs, including miR-34a-5p, miR-34a-3p, miR-34b-5p and miR-34c-5p, were increased in the fibroid compared to the myometrium. The levels of several miRNAs were associated with exposure to particular phthalates, which implies that an altered miRNA expression can be a part of the executive pathway of phthalates. Among the observed associations, a member of the miR-34/449 family miR-449b showed a positive correlation with mono-carboxy-isooctyl phthalate levels [71]. However, additional research needs to be conducted to better understand the link between phthalate exposure, miR-34a and the female reproductive system.

Little is known about the effect of phthalates on miR-34 expression in other tissues. However, it was demonstrated that the prenatal and perinatal exposure of rats to a phthalate mixture causes a decrease in miR-34a expression in the hippocampus during adulthood in female, but not male, rats. Therefore, a phthalate-induced inhibition of miR-34a expression can be considered a long-term and sex-dependent effect [56].

It has been proposed that phthalates are among the obesogenic environmental factors [82]. It is possible that miR-34a is involved in this process, as BBP-treated preadipocytes 3T3-Li exerted an increase in adipogenesis and miR-34a expression. BBP administration also caused a decrease in the expression of miR-34a target genes Nampt and Sirt1. The administration of a miR-34a inhibitor prevented a BBP-induced increase in oil accumulation in 3T3-Li cells and a decrease in the Nampt expression and Sirt1 activity [83]. These results are in accordance with previously reported obesogenic effects of miR-34a [84] and known functions of Nampt and Sirt1 in adipogenesis [85,86]. Therefore, the obesogenic effects of miR-34a are likely to be amplified via interaction with phthalates.

Although research on the pathological effects of phthalates on living organisms has been intensive, it remains problematic due to the ambiguity of dosage-, tissue-specific effects and the cumulative effects of phthalates, depending on the studied mixture of substances. However, as epigenetic regulation based on interactions of phthalates with the miR-34 expression and the consequent impact on the reproductive tract and brain development, as well as oncogenesis and adipogenesis having been convincingly demonstrated, we suppose that the involvement of miRNAs in the execution of phthalates’ effects deserves more attention in the future. 

### 2.2. Interference of an Electromagnetic Field on miR-34-Mediated Regulation

Probably the most problematic of the new environmental factors affecting inhabitants of highly urbanised conglomerations are electromagnetic fields (EMFs). Difficulties in the investigations on EMF-induced physiological effects issue from the variety of wavelengths, intensity and duration of EMF exposure influencing humans and animals. Moreover, the quantification of EMF exposure is not trivial, and there are no simple and affordable devices that could be used for the quantification of cumulative exposure to EMF on an everyday basis. 

In addition to EMFs present in the environment, patients can, in the near future, be exposed to electromagnetic therapy, which seems to be a helpful tool for the targeted delivery of miRNAs to the site of intended action. Artificial molecules of miR-34a packed with iron oxide magnetic nanoparticles were successfully tested for the efficient silencing of programmed death-ligand 1 (PD-L1) to non-small cell lung carcinoma/NSCLC (A549) and triple-negative breast cancer/TNBC (MDA-MB-231) cells [87]. PD-L1 promotes the programmed cell death of T cells present in the tumour microenvironment that result in an immunotolerance to the cancer cells [88]. miR-34a delivered to cells using iron oxide magnetic nanoparticles and magneto targeting successfully decreases the expression of PD-L1 and in this way decreases the vulnerability of the immune system to cancer tissue signalling. Because of this and other promising results [89], the magneto targeting of miRNA directly into the site of intended miRNA effect is a promising therapeutic aim of translational medicine [90]. 

However, there is growing evidence that the expression of some miRNAs can be modulated using EMFs, although the mechanisms of this interaction are far from understood [91,92,93]. Therefore, the use of EMFs to target miRNAs opens question of the possible side effects of EMF on miRNA-mediated actions.

It was demonstrated that extremely low-frequency electromagnetic radiation (ELF-MF, 50 Hz, 1 mT) inhibits the expressions of miR-34b and miR-34c through the repression of its pre-miRNA in a time-dependent manner in both proliferating and differentiated dopaminergic human neuroblastoma cells SH-SY5Y. Transcriptional repression is driven by the hyper-methylation of the miR-34b/c promoter. As ELM-MF also triggers an increase in ROS generation, it was expected that ROS would modulate miR-34b/c levels. However, the administration of free-radical scavengers did not restore the pre-miR-34b/c expression in ELM-MF-exposed cells. Similarly, the ELM-MF-induced inhibition of pre-miR-34b/c expression was not related to a decrease in p53 activity in ELM-EF-treated cells. The EMF-MF-induced expression of the miR-34b target gene synuclein alpha (*snca*) was associated with the incidence of Parkinson’s disease (PD). Therefore, miR-34b seems to interfere with EMF-MF in the regulation of the *snca* expression and possibly PD progression, with miR-34 playing a beneficial role for the patient [94].

An altered functionality of miR-34a in the colorectal cancer cell line DLD1 exposed to EMF with 2.4 GHz frequency (RF-EMF) was demonstrated by our group [36]. The exposure of DLD1 cells to RF-EMF potentiates the capacity of miR-34a to induce the expression of the clock gene *cry1* that was not observed under control conditions. The oncogenic capacity is usually attributed to CRY1 protein [95,96]; therefore, the tumour suppressor capacity of miR-34a seems to be inverted under the conditions of RF-EMF into oncogenic potential. On the other hand, the decrease in the *per2* (which through the inhibition of MDM2 prolongs p53 activity, Figure 1) and *survivin* expression was weakened and abolished, respectively, when miR-34a-transfected cells were exposed to RF-EMF. The capacity of miR-34a to inhibit DLD1 cell metabolism was also lost under the conditions of RF-EMF, which contributes to the implicated decrease in the oncostatic capacity of miR-34a in RF-EMF-exposed cells [36]. 

We are aware that experiments performed with the use of cell culture models have their limits when the results are to be extrapolated to a complex living organism. However, the effects of 2.4 GHz EMF on the expression of components of the circadian transcriptional–translational regulatory loop were so pronounced that the effects of the LD cycle disruption and deregulated circadian rhythms on miR-34 functionality have also been included in this review.

### 2.3. miR-34 Signalling and Its Interaction with the Circadian System

The circadian system governs biological rhythms with a period close to 24 h, and its hierarchical structure is composed of central and peripheral oscillators. The central oscillator is situated in the suprachiasmatic nuclei of the hypothalamus (SCN), localised above the optic nerve chiasm. Peripheral oscillators are present in all other tissues of the human body. Central and peripheral oscillators are interconnected by a complex network of humoral and neural pathways. The localization of the central oscillator just above the optic chiasm is of huge physiological importance as there is a tiny neural pathway connecting the retina and suprachiasmatic nuclei escaping from the optic nerve that provides information about the external LD cycle directly to the SCN without modulatory influences of the thalamus and occipital cortex [97,98].

Both central and peripheral oscillators consist of unicellular oscillators. The central oscillator differs from peripheral tissues mainly by much stronger couplings between its unicellular components compared to the periphery [99]. In both cases, rhythmicity generation is based on the periodically fluctuating expression of clock genes. In humans, there are three homologues of the *period* gene (*per1*, *per2* and *per3*) and two homologues of the *cry* gene (*cry1* and *cry2*). The expression of clock genes is induced by a heterodimer containing the transcription factors BMAL1 and CLOCK, or alternatively by BMAL1 and NPAS2, which is a functional homologue of CLOCK. When the concentrations of protein products of clock genes *per* and *cry* achieve a critical concentration in the cytoplasm, they generate negative transcriptional feedback. PER:CRY heterodimers are translocated to the nucleus, interfere with the BMAL1:CLOCK(NPAS2)-mediated regulation and inhibit *per* and *cry* gene transcription. This sequence of events, called the basic molecular feedback loop of the circadian oscillator, is further influenced by other regulatory factors, e.g., regulation mediated by REV-ERB and ROR via the RORE regulatory domain [100]. Moreover, clock gene expression is modified post-transcriptionally [101]. These processes can influence how the circadian system responds to the external LD cycle and other synchronizing cues [102].

It was convincingly demonstrated that miRNAs contribute to the regulation of the clock gene expression [103,104,105,106] and the generation of circadian oscillations [107,108,109,110]. 

Our group has previously demonstrated that the expression of miR-34a effectively inhibits the expression of *per2* in the colorectal cancer cell lines DLD1 and LoVo [36,37]. Moreover, the expression of *per2* was negatively associated with the expression of miR-34a in tumours of patients undergoing surgery for colorectal cancer treatment at a more advanced stage of disease [111]. An inhibitory effect of ectopic miR-34a on the *per1* expression has been shown in the cholangiocarcinoma cell line Mz-ChA-1. Decrease in the proliferation and invasion of Mz-ChA-1 was associated with the administration of anti-miR-34a, while *per1* over-expression induced apoptosis and caused a decrease in the Mz-ChA-1 proliferation and tumour growth in a xenograft model [112]. However, the expression of *per* genes in humans is not regulated exclusively by miR-34a. The whole *per* family is responsive to regulation mediated by the miR-192/194 cluster [113], and the expressions of *per1* and *per2* are inhibited by miR-24 and miR-29 and modulated by miR-30 [108,109]. miRNA-mediated control of other clock genes is reviewed in detail elsewhere e.g., [105].

An expanding amount of evidence implicates a reciprocal regulatory relationship between the circadian system and miRNA-mediated regulation, which means that the circadian system also regulates the expression of miRNAs [101,105,108,110,114]. It has been shown that the presence of daily rhythms in the expression of miRNA is conserved from plants to mammals [115]. The circadian cis-element E-Box [107,116,117] and RORE [118] have been detected in the upstream regions of the miRNAs exerting the circadian rhythm.

In some cases a simple bidirectional relationship between clock genes and clock-regulated miRNA has been identified [119], e.g., the expression of *bmal1* via a specific binding site in the 3′-UTR region is inhibited by miR-142-3p, and this miRNA exerts a pronounced daily rhythm in its expression in the SCN, which is induced through a conserved canonical E-box [116,117]. 

In the case of miR-34a, the regulatory network of expression is more complex. A distinct daily rhythm in pre-miR-34a has been detected in the liver [110] and prefrontal cortex (PFC) [120], implicating a role of the circadian oscillator in the generation of the miR-34a daily expression pattern. On the other hand, the expression of pre-miR-34a did not exert a daily rhythm in the SCN, heart and kidney [110]. Interestingly, mature and pre-mature forms of mRNA show different patterns in the PFC. Rhythmicity in levels of the mature form of miR-34a emerged only in animals provided with a food reward [120]. 

Different 24 h patterns in the expressions of the precursor and mature forms of miRNA are not rare [107,114,118]. For miRNA, target- and tissue-specific turnover have been implicated as reasons for this phenomenon [114,121]. Our results extend the current knowledge about miRNA turnover, implying that the function executed by a particular tissue can also influence mature miRNA levels [120].

To further elucidate the regulatory relationship between miRNA biosynthesis and the circadian system, we analysed the expression of key enzymes involved in miRNA generation during a 24 h cycle in central and peripheral oscillators. We did not detect a significant daily rhythm in the expression of Dicer and Drosha in the central and peripheral oscillators [110]. However, the mRNA of DGCR8, which together with Drosha cleaves pri-miRNA to form pre-miRNA in the nucleus [2], exerted clear-cut rhythms in the liver, heart and kidney [110]. Interestingly, the expression of *dgcr8* mRNA did not show a daily rhythm in the SCN, and in peripheral tissues, *dgcr8* mRNA rhythmicity differed with respect to the time of maximal expression. 

The disruption of circadian rhythms is frequently observed in humans exposed to shift work, jet lag and/or irregular food intake and has been associated with neuronal imbalance causing sleep disturbances, the occurrence of metabolic syndrome, an increased incidence of some types of cancer and other symptoms [122,123,124,125,126]. miR-34a, as it targets the expression of the clock gene *per2* [36,37] and influences neural stem cell proliferation and differentiation [127,128] was implicated in the neuronal response to confusing synchronizing signals. It has been demonstrated that exposing rats to dim light at night (dLAN) for three weeks significantly decreases miR-34a expression in the hippocampus [129]. The presence of miR-34 family members is necessary for the generation of stress-related behavioural responses [130]. Therefore, it is implied that a decrease in the miR-34a expression in response to dLAN exposure contributes to dLAN-induced changes in behavioural parameters and the impairment of cognitive functions observed in dLAN-exposed rats [129]. 

## 3. Conclusions

The main aim of the present review was to demonstrate that environmental factors possess the capacity to influence miRNA-mediated regulation and that these effects can be cumulative (Figure 2). Because of the well-known physiological effects of the miR-34 family, it was used as a miRNA example.

To elucidate the molecular pathways involved in the interferences between the miR-34 family and environmental factors, we performed an in silico analysis, of which its details are provided in the legend of Figure 3 and in Appendix A. A PANTHER Pathway analysis of miR-34 regulated genes revealed that miR-34 signalling is mostly involved in the regulation of cell proliferation, neural functions and reproduction (Figure 3A). Similarly, genes that are under control of the miR-34 family members and are responsive to phthalates administration or electromagnetic field exposure are involved in the regulation of all above-mentioned physiological processes (Figure 3C,D; respectively). These data are in accordance with resources listed in Table 1.

Based on the available experimental evidence, we conclude that the exposure of organisms to phthalates, EMFs and circadian disruption significantly influences miR-34a functioning with respect to cancer progression, neuroplasticity and reproduction (Table 1, Figure 3). Especially in the case of miR-34-modulated cancer progression and neuroplasticity, a regulatory interference of phthalates, EMFs and a disrupted LD cycle is inevitable. Unfortunately, this set of circumstances is very difficult to approach, both experimentally and epidemiologically. Therefore, the synthesis of information about the interference of particular factors and specific miRNAs could reveal how the environment influences miRNA signalling. As exposure to the factors that were in the focus of the present review is long-lasting and difficult to control, we suppose that their specific as well as their cumulative impact on miRNA function should be addressed in more detail, especially with respect to translational medicine focused on gene silencing using miRNAs and public health oriented research.

## Figures and Tables

**Figure 1 biomedicines-12-00424-f001:**
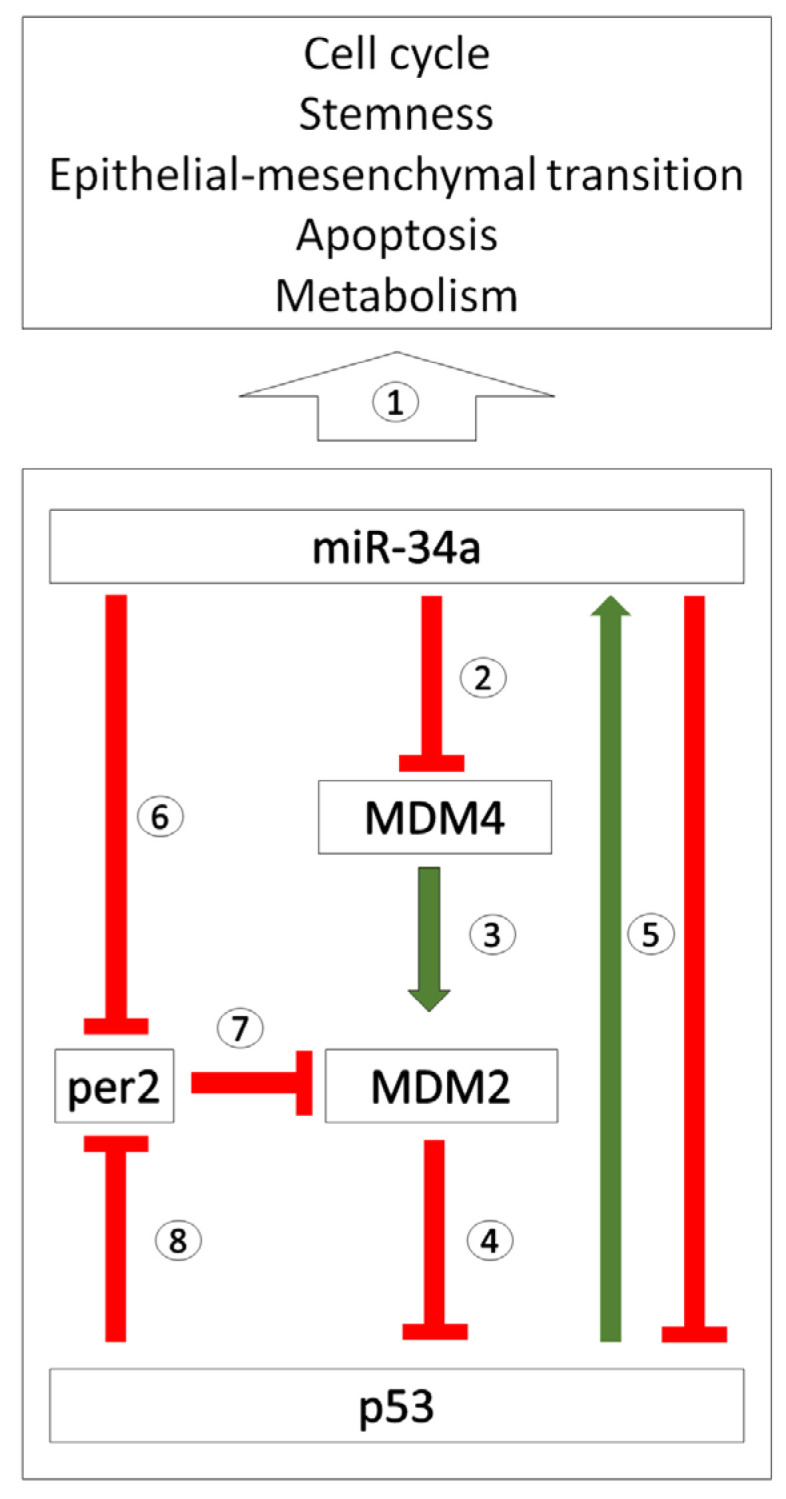
Scheme of the regulatory network between miR-34a, MDM2 and p53. The expression of the MDM4 regulator of p53 (MDM4) is inhibited by miR-34a. As MDM4 is a positive regulator of MDM4, miR-34a contributes to MDM2 proto-oncogene (MDM2) activity through the inhibition of MDM4; miR-34a prolongs the activity of p53. On the other hand, miR-34a targets p53 and inhibits its expression. p53 induces the expression of miR-34a. miR-34a-5p also inhibits the expression of clock gene *per2*, which is an inhibitor of MDM2 activity. p53 inhibits the expression of *per2*. Circled numbers implicate references supporting particular links in the scheme. 1—[30,31,32]; 2—[28,33]; 3—[28,33]; 4—[28,33]; 5—[27,34,35]; 6—[36,37]; 7—[38,39]; 8—[40].

**Figure 2 biomedicines-12-00424-f002:**
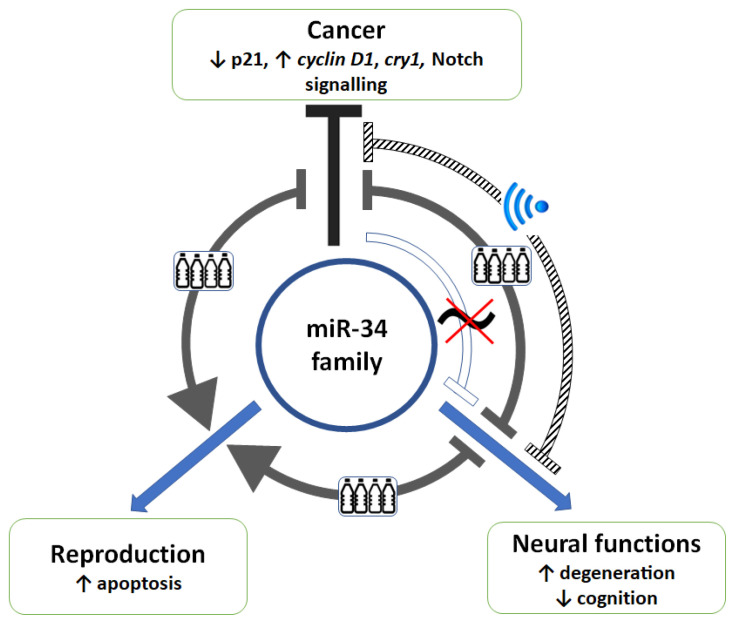
Cumulative effects of an electromagnetic field (EMF, crosshatch lines), phthalates (grey lines) and a deregulated circadian system (white lines) on regulation mediated by miR-34 family members. *Neural functions:* EMF exposure, which causes a decrease in the miR-34b/c expression in SH-SY5Y and primary cortical neuron cells, also predisposes neural cells to degeneration. This statement was made as the expression of the miR-34b target gene synuclein alpha increased, and the expression of miR-34b decreased in SH-SY5Y cells affected by the EMF, implicating neuronal degeneration under these conditions [94]. The deregulation of the circadian system induced by dim light at night caused a significant decrease in the miR-34a expression, which was associated with the impairment of cognitive functions in rats [129]. The precursor form of miR-34a exerts a circadian rhythm in the prefrontal cortex of rats. The daily rhythm in mature miR-34a after a pronounced phase shift was observed only in animals whose entrainment was strengthened by a regular provision of a food reward. The role of miR-34a has been implicated in dopaminergic signalling [120]. *Reproduction:* With respect to reproduction, an interaction of phthalates and miR-34b-mediated signalling has been observed in the testes of mice exposed to endocrine-disrupting chemicals (EDCs) for 60 days. EDCs induced the apoptosis of germ cells, which was associated with an increased expression of miR-34b [73]. *Cancer progression:* The administration of miR-34a significantly decreased the expression of clock genes *per2* and *bmal1* [36] and increased that of *clock* [37] in DLD1 and/or LoVo colorectal cell lines. The EMF modified the influence of miR-34a on the circadian oscillator [36]. The EMF exposure of DLD1 cells transfected with miR-34a caused an increase in the expression of the clock gene *cryptochrome 1* (*cry1*) with oncogenic properties. This effect was not observed under control conditions. On the other hand, the inhibition of the survivin expression, which, under control conditions, was observed in response to miR-34a administration, disappeared when cells were exposed to the EMF. Therefore, it was hypothesised that EMF changes the oncostatic potential of miR-34a [36]. Phthalate administration caused a decrease in miR-34a and p21 and an increase in the cyclin D1 expression in LNCaP and PC-3 cells, which was associated with increased cell viability [76]. Accordingly, phthalate administration caused a decrease in the miR-34a levels in the rat liver and an increase in Notch signalling. The change in Notch signalling was attributed to the absence of an inhibitory effect of miR-34a on the *Notch1* expression [77].

**Figure 3 biomedicines-12-00424-f003:**
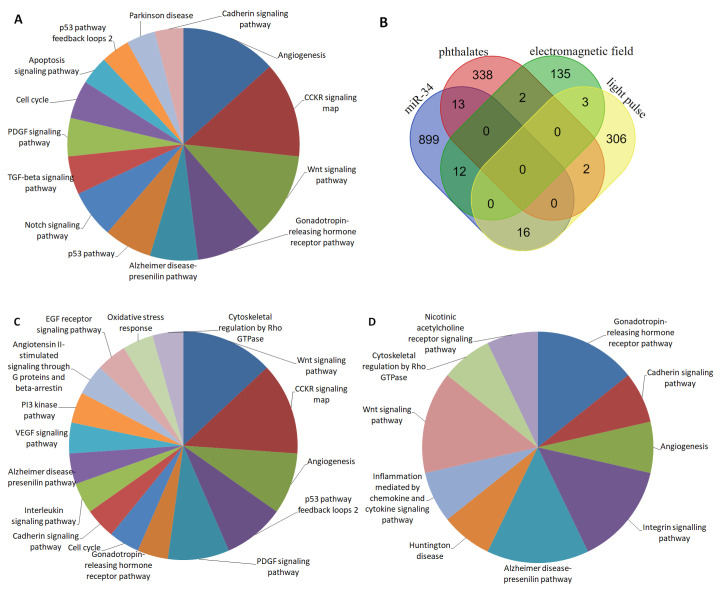
Analysis of pathway involved in miR-34-mediated signalling. (**A**) PANTHER Pathway analysis [131,132] of genes of which their connection with miR-34 is supported by strong evidence according to the miRTarBase [20]. Only PANTHER categories containing three and more genes are shown. (**B**) Vein diagram showing overlap between genes regulated by miR-34 family members and particular environmental factor. All genes referred in respect to miR-34 in the miRTarBase were used in this search. The references of papers containing datasets of genes induced by environmental factors are provided below. (**C**) PANTHER Pathway analysis of genes under the control of miR-34 that are responsive to phthalate administration [133,134,135]. (**D**) Pathway analysis of genes under the control of miR-34 that are responsive to electromagnetic field exposure [136,137,138]. A pathway analysis of genes controlled by miR-34 and influenced by bright light [139] is not show as most of the genes are not attributed to a particular PANTHER Pathway.

**Table 1 biomedicines-12-00424-t001:** Changes in the miR-34 expression in response to an electromagnetic field, phthalate administration and circadian disruption.

Stress Factor	*miRNA*	Change in Expression	Tissue/Cell Line (Species)	Reference
Phtalates	*pre-miR-34b* *miR-34b-5p*	up	Testes (mouse)	[73]
	*miR-34a-5p*	up	3T3-L1 cells (mouse)	[83]
	*miR-34a*	down	PC-3 cells, LNCaP cells (human)	[76]
	*miR-34a-5p*	down	Female/CA3 and DG in Hippocampus (rat)	[56]
	*miR-34a-5p*	down	Liver (rat)	[77]
EMF	*miR-34b/c*	down	SH-SY5Y cells (human) PCN cells (mouse)	[94]
circadian cycle deregulation/ cancer	*pre-miR-34a*	up	Kidney, Heart (rat)	[110]
	*miR-34a-5p*	up	Colorectal cancer tissue (human)	[111]
	*miR-34a-5p*	aberrant	Cholangeocarcinoma cells (human)	[112]
circadian cycle deregulation/neurodegeneration	*miR-34a*	down	Hippocampus (mouse)	[129]

EMF—electromagnetic field, DG—dentate gyrus, PCN—primary cortical neurons.

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
