# Peer review of "Impact of Long-Lasting Environmental Factors on Regulation Mediated by the miR-34 Family"

_biomedicines, 2024, doi:10.3390/biomedicines12020424_

Round 1

Reviewer 1 Report

Comments and Suggestions for Authors

In this manuscript, the authors have provided a collective information on the regulation of miR-34 by long-lasting environmental factors. This is an interesting topic. The following points should be addressed to improve the manuscript before considering for the publication.

1. A separate section on the therapeutic significance of miR-34 should be included.

2.  An additional figure on the miR-34 signaling networks will improve the review manuscript. 

Comments on the Quality of English Language

Moderate English language editing needed.

Author Response

Dear reviewer,

Thank you very much for your valuable time and suggestions and advice. Your suggestions and comments were implemented into the text, please see detailed response bellow.

We hope that text of MS was improved in such a way that it is acceptable for publication now.

With many thanks and best regards,

the authors

In this manuscript, the authors have provided a collective information on the regulation of miR-34 by long-lasting environmental factors. This is an interesting topic. The following points should be addressed to improve the manuscript before considering for the publication.

  1. A separate section on the therapeutic significance of miR-34 should be included.

- separate section focused on ways how targeting of miR-34a can be improved and which type of cancers could be treated in this way in future was incorporated in the Chapter 2 and is labelled with yellow colour.

  1. An additional figure on the miR-34 signalling networks will improve the review manuscript. 

- additional figure (figure 3) displaying miR-34a-signalling network was created and incorporated in the chapter “Conclusions”. To show the most up to date information we performed search in database miRTarBase and exported all target genes of miR-34 family members. Consequently we extracted those genes that were supported by “strong evidence” in miRTarBase and performed PANTHER Pathway analysis to reveal which molecular pathways are involved in miR-34 mediated signalling.

Next we performed literature search and found papers containing datasets of genes changed in response to phthalates administration and/or electromagnetic field exposure and/or bright light pulse and calculated Venn diagram to see overlap between miR-34 and particular environmental factor induced signalling.

As expected, there was an overlap in all cases. We performed PANTHER Pathway analysis for genes that were induced by miR-34 and each particular environmental factor. Results are shown in figure 3. Text describing figure was included in the chapter “Conclusions”.

We hope that in this way we provided sufficient information about molecular pathways induced by miR-34 not only in control condition but also under challenging environmental circumstances.

References from which gene expression datasets were obtained are provided in the figure 3 legend. List of genes used in analysis and details of PANTHER Pathway analysis are provided in Supplementary file 1.

Reviewer 2 Report

Comments and Suggestions for Authors

Comments:

1. On Table 1: Include species as a designated category. If Table 1 features only one particular species, modify the title to reflect that, such as "....... family in mice."

2. In Figure 1, authors should incorporate information about the impact of p53 on the miR-34 family-mediated effects.

3. What about MDM2 in miR-34 family-mediated effect?

Author Response

Dear reviewer,

Thank you very much for your valuable time and suggestions, we appreciate them very much. All your suggestions were implemented into the text. We hope that you will find revised MS acceptable for publication.

With many thanks and best regards,

the authors

  1. On Table 1: Include species as a designated category. If Table 1 features only one particular species, modify the title to reflect that, such as "....... family in mice."

- thank you for the suggestion, Table 1 was changed as requested

  1. In Figure 1, authors should incorporate information about the impact of p53 on the miR-34 family-mediated effects.

- dear reviewer, of course we strongly agree that p53 induces expression of all members of miR-34 family (He et al., 2007), miR-34a indirectly increases p53 activity (Okada et al., 2014) and that p53 contributes to manifestation of miR-34 family a lot. However, there is also experimental evidence implicating p53 independent effect of miR-34 members (e.g. Slabakova et al., 2017; Olejarova et al., 2022; Moravčík et al., 2023).

Former figure 1 showed only those regulatory relationships between miR-34 family members and physiological function that were experimentally proved. As recently there is no experimental evidence interconnecting changes in miR-34 and p53 levels and/or functioning explicitly under conditions of light:dark regimen disruption and/or electromagnetic field exposure and/or after phthalates administration, we could not add p53 between links of former figure 1 in spite the fact that its involvement is very likely.

Instead of incorporation of p53 into former figure 1 we created new figure (in recent MS referred as figure 1) and incorporated it in the Chapter 2. The new figure 1 displays regulatory relationships between p53, miR-34a, MDM2 and MDM4 and clock gene per2.

  1. What about MDM2 in miR-34 family-mediated effect?

- of course, we agree that MDM2 is critical in regulation of p53 activity as it targets p53 for degradation. Since MDM4 does not possess ubiquitin ligase activity and cannot bind to p53 it regulates p53 activity indirectly, via interaction with MDM2. MDM4 stabilises MDM2 and in this way is promotes p53 degradation (Okada et al., 2014; Cao et al., 2020). p53 and miR-34 exert regulatory interconnection. p53 induces miR-34a expression and miR-34a indirectly induces p53 activity by silencing MDM4 and sirtuin 1 (Okada et al., 2014; He et al., 2007).

There is no evidence in databases miRTarBase https://mirtarbase.cuhk.edu.cn/~miRTarBase/miRTarBase_2022/php/index.php or  TargetScanHuman https://www.targetscan.org/vert_80/ that MDM2 is targeted by leading strand of some member of miR-34 family. Similarly, bases on the literature search, we did not reveal evidence about direct MDM2 inhibition executed by some miR-34 family member.

Recent knowledge about interconnections between miR-34 family and p53 is summarized in the new figure 1 implemented in the Chapter 2.

Cao L, Liu Y, Lu JB, Miao Y, Du XY, Wang R, Yang H, Xu W, Li JY, Fan L. A feedback circuit of miR-34a/MDM4/p53 regulates apoptosis in chronic lymphocytic leukemia cells. Transl Cancer Res. 2020; 9(10):6143-6153. doi: 10.21037/tcr-20-1710.

He L, He X, Lim LP, de Stanchina E, Xuan Z, Liang Y, Xue W, Zender L, Magnus J, Ridzon D, Jackson AL, Linsley PS, Chen C, Lowe SW, Cleary MA, Hannon GJ. A microRNA component of the p53 tumour suppressor network. Nature. 2007; 447(7148):1130-4. doi: 10.1038/nature05939.

Moravčík R, Olejárová S, Zlacká J, Herichová I. Effect of miR-34a on the expression of clock and clock-controlled genes in DLD1 and Lovo human cancer cells with different backgrounds with respect to p53 functionality and 17β-estradiol-mediated regulation. PLoS ONE 2023; 18(10):e0292880. https://doi.org/10.1371/journal.pone.0292880

Okada N, Lin CP, Ribeiro MC, Biton A, Lai G, He X, Bu P, Vogel H, Jablons DM, Keller AC, Wilkinson JE, He B, Speed TP, He L. A positive feedback between p53 and miR-34 miRNAs mediates tumor suppression. Genes Dev. 2014;28(5):438-50. doi: 10.1101/gad.233585.113.

Olejárová S, Moravčík R, Herichová I. 2.4 GHz Electromagnetic Field Influences the Response of the Circadian Oscillator in the Colorectal Cancer Cell Line DLD1 to miR-34a-Mediated Regulation. Int J Mol Sci. 2022; 23(21):13210. doi: 10.3390/ijms232113210.

Slabáková E, Culig Z, Remšík J, Souček K. Alternative mechanisms of miR-34a regulation in cancer. Cell Death Dis. 2017; 8(10):e3100. doi: 10.1038/cddis.2017.495.

Round 2

Reviewer 2 Report

Comments and Suggestions for Authors

No mre comments